# Advancements in Diagnosis of Neoplastic and Inflammatory Skin Diseases: Old and Emerging Approaches

**DOI:** 10.3390/diagnostics15162100

**Published:** 2025-08-20

**Authors:** Serena Federico, Fortunato Cassalia, Marcodomenico Mazza, Paolo Del Fiore, Nuria Ferrera, Josep Malvehy, Irma Trilli, Ana Claudia Rivas, Gerardo Cazzato, Giuseppe Ingravallo, Marco Ardigò, Francesco Piscazzi

**Affiliations:** 1Dermatology Unit, Department of Health Sciences, Magna Graecia University, 88100 Catanzaro, Italy; drserenafederico@gmail.com; 2Dermatology Unit, Department of Medicine (DIMED), University of Padua, 35121 Padua, Italy; fortunato1287@gmail.com; 3Soft-Tissue, Peritoneum and Melanoma Surgical Oncology Unit, Veneto Institute of Oncology IOV-IRCSS, 35121 Padova, Italy; marcodomenico.mazza@iov.veneto.it (M.M.); paolo.delfiore@iov.veneto.it (P.D.F.); 4Dermatology Department, Hospital Clínic Barcelona, 08036 Barcelona, Spain; ferrera@recerca.clinic.cat (N.F.); jmalvehy@clinic.cat (J.M.); acrivas@recerca.clinic.cat (A.C.R.); 5Fundació Clínic per a la Recerca Biomèdica (FCRB), Instituto de Investigaciones Biomédicas August Pi i Sunyer (IDIBAPS), 08036 Barcelona, Spain; 6Instituto Carlos III, CIBER de Enfermedades Raras, 08036 Barcelona, Spain; 7Department of Interdisciplinary Medicine, University of Bari “Aldo Moro”, 70124 Bari, Italy; trilliirma@gmail.com; 8Pathology Unit, Department of Precision and Regenerative Medicine and Ionian Area, University of Bari “Aldo Moro”, Piazza Giulio Cesare 11, 70121 Bari, Italy; giuseppe.ingravallo@uniba.it; 9Dermatology Unit, IRCCS Humanitas Research Hospital, Rozzano, 20089 Milano, Italy; marco.ardigo@hunimed.eu (M.A.); francesco.piscazzi@humanitas.it (F.P.); 10Department of Biomedical Sciences, Humanitas University, Pieve Emanuele, 20072 Milano, Italy

**Keywords:** dermatologic diagnosis, dermoscopy, confocal microscopy, artificial intelligence, skin cancer, inflammatory skin diseases, line-field OCT, digital health

## Abstract

**Background:** In recent decades, dermatological diagnostics have undergone a profound transformation, driven by the integration of new technologies alongside traditional methods. Classic techniques such as the Tzanck smear, potassium hydroxide (KOH) preparation, and Wood’s lamp examination remain fundamental in everyday clinical practice due to their simplicity, speed, and accessibility. At the same time, the development of non-invasive imaging technologies and the application of artificial intelligence (AI) have opened new frontiers in the early detection and monitoring of both neoplastic and inflammatory skin diseases. **Methods:** This review aims to provide a comprehensive overview of how conventional and emerging diagnostic tools can be integrated into dermatologic practice. **Results:** We examined a broad spectrum of diagnostic methods currently used in dermatology, ranging from traditional techniques to advanced approaches such as digital dermoscopy, reflectance confocal microscopy (RCM), optical coherence tomography (OCT), line-field confocal OCT (LC-OCT), 3D total body imaging systems with AI integration, mobile applications, electrical impedance spectroscopy (EIS), and multispectral imaging. Each method is discussed in terms of diagnostic accuracy, clinical applications, and potential limitations. While traditional methods continue to play a crucial role—especially in resource-limited settings or for immediate bedside decision-making—modern tools significantly enhance diagnostic precision. Dermoscopy and its digital evolution have improved the accuracy of melanoma and basal cell carcinoma detection. RCM and LC-OCT allow near-histological visualization of skin structures, reducing the need for invasive procedures. AI-powered platforms support lesion tracking and risk stratification, though their routine implementation requires further clinical validation and regulatory oversight. Tools like EIS and multispectral imaging may offer additional value in diagnostically challenging cases. An effective diagnostic approach in dermatology should rely on a thoughtful combination of methods, selected based on clinical suspicion and guided by Bayesian reasoning. **Conclusions:** Rather than replacing traditional tools, advanced technologies should complement them—optimizing diagnostic accuracy, improving patient outcomes, and supporting more individualized, evidence-based care.

## 1. Introduction

Diagnostic techniques in dermatology have undergone significant advancements over the past few decades, and, although dermatological diagnosis has relied primarily on clinical examination, the support of histopathological analysis is mandatory in different cases [1]. However, histological examination of skin biopsy samples is not always feasible and may not be well tolerated by patients, and this has led to the need for alternative diagnostic methods capable of providing acceptable diagnostic accuracy for both oncologic and inflammatory skin diseases [2]. In this regard, dermoscopy, once considered a complementary tool, is now essential for early detection and clinical classification of pigmented and non-pigmented lesions, especially melanoma [2]. Indeed, an increasing number of detailed dermoscopic patterns are being studied and categorized into sub-classifications based on different anatomical locations. For instance, acral lesions exhibit specific suspicious dermoscopic patterns that differ from those found in other body sites [1,2]. Furthermore, new dermoscopic classifications are continuously emerging, providing clinicians with valuable support in diagnostic decision-making, and the addition of technique like reflectance confocal microscopy (RCM) and line-field confocal optical coherence tomography (LC-OCT) allows non-invasive visualization of skin structures, improving diagnosis of neoplasms such as basal cell carcinoma (BCC), actinic keratosis (AK), and others [3]. On the other hand, the identification of novel molecular and genetic biomarkers has enabled early diagnosis and enhanced both primary and secondary prevention strategies in the dermatological field, such represented by BIOMAP (biomarkers in atopic dermatitis and psoriasis) [4].

Artificial intelligence has played a pivotal role in standardizing care by enabling the development of platforms containing medical images and easily accessible diagnostic-therapeutic algorithms [5]. Nonetheless, despite the vast amount of accessible and shareable information, clinical experience and the physician’s intuition grounded in bedside anamnesis remain indispensable for establishing an effective diagnostic therapeutic pathway.

Basic diagnostic techniques remain a cornerstone in dermatology and dermatopathology, and, in particular, histological examination, enhanced by immunohistochemical staining methods, continues to represent the primary diagnostic tool, enabling through direct comparison the advancement and systematization of the most recent non-invasive diagnostic techniques mentioned above [1,6].

This review aims to explore the evolution of both invasive and non-invasive dermatological diagnostics, examining how emerging and advanced diagnostic techniques integrate with conventional methods in the clinical diagnosis of skin diseases.

## 2. Results

### 2.1. Traditional Dermatologic Diagnostic Methods

Although scientific innovations are revolutionizing diagnostics in dermatology, traditional diagnostic techniques remain firmly anchored in daily clinical practice [1]. These well-established methods, supported by much scientific evidence, have a significant diagnostic value for factors such as simplicity of application, availability, and immediacy of response [1,6]. The Tzanck smear, microscopy with potassium hydroxide (KOH), skin preparations derived from scraping of the skin for scabies, and Wood’s lamp examination are among the main and most used methods. These techniques simplify differential diagnosis in a number of clinical conditions, such as pityriasis versicolor, a condition that should be distinguished from hypopigmentation due to melanocyte destruction as in vitiligo. Wood’s lamp examination helps in this differentiation based on the type of fluorescence emitted; indeed, a white-milky fluorescence is indicative of vitiligo, while a yellowish or orange fluorescence suggests a fungal infection [7]. The clinical relevance of these traditional techniques lies in their ease of use, and in an era increasingly dominated by complex technologies, these basic diagnostic tools should be integrated with emerging methodologies and serve as tools for validating the most innovative techniques [7].

The Tzanck smear, a technique devised by Arnault Tzanck in 1947, is a cytological diagnostic technique used to assess various types of skin lesions, including vesicles, blisters, and pustules. The procedure involves the collection of cells by scraping at the base of a skin lesion, followed by staining with Giemsa, Wright, or Papanicolaou to enable optical microscope vision [7,8]. The test aims to identify multinucleate giant cells and acantholytic keratinocytes, cytological characteristics suggestive of infections caused by herpes simplex virus (HSV), varicella zoster virus (VZV), or autoimmune diseases such as pemphigus vulgaris and other dermatosis. The sensitivity of the Tzanck smear to detect herpes infections ranges from 73% to 86%, with a specificity of about 85% [8]. Although the polymerase chain reaction (PCR) test offers the highest diagnostic accuracy, the Tzanck smear remains in use, particularly in circumstances where PCR is not readily available. Conventional staining methods do not differentiate between HSV and VZV, and the immunofluorescence is the method of choice as it offers a significantly higher antigen positivity index than either electron microscopy or virus isolation [7,8]. In addition, the Tzanck smear provides a quick and low-cost diagnostic option in both outpatient and hospital settings, and this is particularly critical in clinical settings involving acute blight vesicles, where early diagnosis and immediate clinical decision-making are essentials [9].

The preparation of KOH is a first-line laboratory diagnostic technique widely used for direct identification of superficial fungal infections, including dermatophytosis (tinea), skin candidiasis, and pityriasis versicolor (Figure 1). This method is simple and non-invasive, allowing microscopic observation of fungal elements without the need for preliminary culture steps. One critical but often underestimated aspect is the careful collection of biological material from the lesion site, ensuring sampling both from the central areas and, when necessary, around the lesion. Skin scales can be collected using a sterile blade, prelevating nail fragments in case of suspected onychomycosis or hair torn from the periphery of alopecic lesions. In some forms of tinea capitis, sampling from the center of the lesion can produce false negative results due to the intense inflammatory response triggered by the pathogenic fungus, which hinders the detection of the fungus. Therefore, peripheral sampling is essential for accurate diagnosis. KOH acts as a keratolytic agent, selectively dissolving keratin, hair, and nails, as well as epidermal cellular debris. This digestion process, sometimes facilitated by a slight heating of the slide, facilitates the screening and visualization of the fungal elements under optical microscopy. Depending on the type of infection, various fungal morphologies can be observed: septa, branched hyphae in dermatophytic infections; pseudohyphae and blastospores in candidiasis; and the characteristic appearance “spaghetti e polpette”, short hyphae and round spores, typical of infections by Malassezia furfur (pityriasis versicolor) [7,8,9].

The diagnostic sensitivity of KOH preparation for dermatophytoses generally ranges between 80% and 95%, depending on sample quality and the operator’s expertise, while specificity can exceed 90%, making it a reliable diagnostic tool. A similar technique is employed for diagnosing infestations by *Sarcoptes scabiei*, the causative agent of scabies. In these cases, the sample consists of epidermal scrapings from characteristic sites such as interdigital spaces, wrists, areolae, or genital regions, corresponding to burrows, the pathognomonic lesions, or papules and vesicles. The specimen is also treated with KOH to dissolve the skin matrix, enabling microscopic identification of adult mites, eggs, and fecal pellets (scybala). However, the sensitivity of this test for scabies is more variable, ranging from 40% to 90%, depending on factors such as anatomical site and is highly operator-dependent. Figure 2 shows an example of microscopic examination with KOH preparation for the detection of *Sarcoptes scabiei*.

In cases of infestation with low parasite load or the presence of secondary lesions, KOH preparation remains a crucial tool for the diagnostic confirmation of cutaneous infections and infestations. It is particularly valuable in high-prevalence settings, institutional outbreaks (such as in nursing homes or schools), and pediatric populations, where rapid diagnosis is essential for therapeutic management and controlling potential epidemic outbreaks [10,11].

Perfected for the first time in the 1920s, the Wood lamp examination remains an important diagnostic tool in dermatology [12], using ultraviolet (UV-A) light with a wavelength of 365 nm, capable of emitting fluorescence from some lesions of the skin. In a completely dark environment, this technique allows for the visualization of undetectable skin lesions under natural light, offering a fast diagnostic technique that is non-invasive and well tolerated by the patient [12,13]. One of the most common applications of the lamp is in identifying fungal infections of the scalp, because thanks to the typical fluorescence patterns emitted under UV light, it is possible to distinguish specific species of Microsporum, such as *Microsporum canis* which exhibit a bright green fluorescence pattern. This facilitates easy and immediate diagnosis, with a sensitivity of over 80%, but, in contrast, dermatophytes of the genus Trichophyton are not fluorinated, allowing an effective differentiation of pathogens [13]. Furthermore, Wood’s lamp is also particularly useful for diagnosing erythrasma, a superficial cutaneous infection caused by *Corynebacterium minutissimum*; in this condition, the bacterial porphyrins produce a red coral fluorescence, which is considered to be pathognomonic [14]. Figure 3 shows an example of erythrasma under Wood’s light.

Another important field of application is the assessment of pigmentary disorders, such as vitiligo (that display sharply demarcated lesion borders under UV light), enabling precise monitoring during follow-up and treatment [15]. These classical methods remain indispensable tools not only in clinical practice but also in the training of future dermatologists. Their integration with modern diagnostic techniques enhances the accuracy and effectiveness of therapeutic strategies.

### 2.2. Dermoscopy and Digital Dermoscopy

Dermoscopy is a non-invasive diagnostic technique that enables the magnified visualization of skin structures invisible to the naked eye. Since the description of dermoscopic patterns and the emergence of handheld dermoscopy devices in the 1980s and 1990s, growing interest in this technique and advances in dermoscopic instruments have made it an essential tool in the daily clinical practice of dermatologists, particularly aiding in the decision to excise a potentially malignant lesion.

There are several dermoscopy modalities, which can be broadly categorized into analog and digital systems. Analog dermoscopy refers to traditional optical devices without digital capture, allowing direct visualization of skin structures under visible light, using various LED illumination sources, and may employ polarized or non-polarized light, with or without skin contact [16]. In non-polarized dermoscopy, the use of immersion fluids is common to reduce surface reflection. The incorporation of ultraviolet (UV) light has further expanded diagnostic capabilities. UV dermoscopy enhances contrast in pigmented lesions, improves tumor margin delineation, and enables real-time visualization of fluorescence in bacterial and fungal infections, porphyrias, and pigmentary disorders [17].

In contrast, digital dermoscopy (DD) integrates optical systems with digital cameras to allow image acquisition, storage, and longitudinal comparison, thereby supporting the follow-up of individual lesions. The combination of total body photography and sequential DD, called the “two-step method”, facilitates the early detection of malignant lesions while reducing the rate of excisions [18], and currently represents the standard screening approach for high-risk melanoma patients. Contemporary high-definition systems now include quantitative analysis and deep learning algorithms capable of detecting subtle morphological changes and generating differential diagnoses based on large dermatological image databases [19]. In recent years, novel devices combining total body photography with polarized light and non-contact DD have been introduced, demonstrating image quality comparable to conventional dermoscopy, with the added benefit of increased time efficiency [20].

Among the most advanced digital modalities, super-high magnification dermoscopy, capable of reaching up to 400× magnification, allows visualization of cellular-level details. This technique has demonstrated potential for identifying melanocytes with irregular arrangements and atypical morphology in melanomas [21], folliculotropism in facial lentigo maligna [22], vascular patterns in basal cell carcinoma [23], and melanophages in different lesions, such lichen planus-like keratosis [24].

In skin oncology, DD has significantly improved diagnostic accuracy, particularly in the surveillance of patients with multiple nevi or at high risk for melanoma. Meta-analyses consistently support its diagnostic utility. Kittler et al. demonstrated that dermoscopy increases diagnostic accuracy for melanoma by 49% compared to unaided visual inspection [25]. Similarly, Vestergaard et al. reported that dermoscopy achieved a significantly higher sensitivity (90%) than naked-eye examination (71%) in the diagnosis of melanoma [26]. For basal cell carcinoma, dermoscopy also shows high diagnostic performance. The meta-analysis by Reiter et al. showed pooled sensitivity and specificity reaching 91.2% and 95%, respectively [27]. Comparative studies have shown that adding dermoscopy to clinical examination increases sensitivity from 66.9% to 85% and specificity from 97.2% to 98.2% [27]. In squamous cell carcinoma (SCC), dermoscopy improves diagnostic sensitivity by revealing patterns associated with lesion subtypes and stages of progression, facilitating the differentiation between actinic keratosis, Bowen’s disease, and invasive SCC [28].

Beyond oncologic diagnosis, dermoscopy has proven valuable in a variety of non-neoplastic dermatologic conditions. Inflammatory dermatoses such as psoriasis, eczema, and lichen planus exhibit characteristic vascular and scaling patterns under dermoscopic examination [29,30]. Although numerous case reports and descriptive series highlight the potential of dermoscopy in this context, further research is needed to establish standardized diagnostic criteria and to quantify the sensitivity and specificity of dermoscopic features in inflammatory skin diseases. In infectious diseases, dermoscopy identifies pathognomonic signs such as the “jet with contrail” in scabies, with sensitivity over 90% [30,31,32]. Dermoscopy is also widely used in trichology to distinguish between alopecia areata, androgenetic alopecia, and frontal fibrosing alopecia, providing a non-invasive, real-time trichoscopic assessment [31,33,34].

Figure 4 shows an example of pigmented melanocytic lesion analyzed with dermoscopy and optical super high magnification dermoscopy.

### 2.3. Reflectance Confocal Microscopy (RCM)

RCM is a non-invasive imaging technique that enables real-time, high-resolution visualization of the skin at nearly cellular detail. It utilizes a low-power near-infrared laser (typically ~830 nm) and a confocal optical system to capture horizontal optical sections of the skin in vivo [35]. It has lateral resolution of nearly 0.5–1 µm and axial resolution of 3–5 µm which allow it to visualize individual cells (keratinocytes, melanocytes, and leukocytes), mimicking an in vivo histological examination without the need for a biopsy. The system works by focusing light on a single point within the skin and filtering out-of-focus reflections with a pinhole, thereby producing thin optical sections. Image contrast is generated by naturally reflective structures such as melanin, which allows the visualization of nuclei, cellular architecture, and dermal collagen fibers. An RCM reach to 200–300 µm deep is common and adequate to evaluate the epidermis, DEJ, and upper papillary dermis [36]. VivaScope 1500 (VS1500) can process a number of small FOVs by mosaicking into the larger lesional image (up to 2 × 2 mm). For anatomical sites (i.e., face, genitalia, hands, feet) where the VS1500 cannot be placed, a handheld device, the VivaScope 3000 (VS3000), is available with a 500 × 500 µm field of view [37]. RCM has gained recognition for its utility in the evaluation of cutaneous tumors, especially when clinical and dermoscopic findings are inconclusive. In melanocytic lesions, RCM enhances diagnostic specificity for melanoma (Figure 5 RMC) by detecting features such as architectural disarray at the DEJ, pagetoid spread, irregular nests, and cytologic atypia [38]. When operated by expert users, sensitivity has ranged from 92% to 97% and specificity from 70% to 80%. When used in conjunction with dermoscopy, the negative predictive value for melanoma can be nearly 100%, which permits safe clinical observation of benign-appearing lesions and a reduction in the number of superfluous biopsies [39].

RCM is also effective in identifying basal cell carcinoma (BCC) features, including dark tumor islands in hypopigmented variants and bright tumor islands in pigmented BCCs, along with peripheral palisading, cleft-like spaces, and horizontal vessels [40]. While clefts in BCC were historically considered a fixation-related histological artifact, recent studies using in vivo RCM and OCT have shown that these structures correspond to true peritumoral spaces filled with mucin [41]. One recent study shows an impressive correlation between the dimension of the dark cleft seen in RCM and the deposit of mucin seen in histopathology [42]. It is particularly helpful in pre-surgical tumor margin mapping of Mohs surgery, helping define subclinical extension and reducing the number of excision stages [41]. Reported sensitivities and specificities for BCC exceed 90% and 95%, respectively [42]. Although its main application is currently in the diagnosis of melanoma and BCC, RCM has been shown to be useful in recognizing features of other cutaneous tumors such as squamous cell carcinoma [43] and for the differential diagnosis of mucosal lesions [44]. In these situations, RCM can direct biopsies toward the most suspicious areas, thus increasing diagnostic yield. Moreover, RCM is being used more often in the follow-up of treatment response to superficial BCC, particularly, after topical treatments, and for the identification of early tumor recurrence. New applications are for differential diagnosis of inflammatory dermatoses (e.g., differentiation of psoriasis versus eczema) or monitoring of inflammatory disease activity [45].

Despite its benefits, RCM has limitations. Its limited penetration depth may be insufficient for evaluating deeper dermal structures, such as in nodular melanomas or invasive squamous cell carcinomas. The field of view is relatively small (typically 0.5 to 1 mm^2^), meaning that larger lesions require time-intensive mosaicking, potentially impacting clinical workflow. Comprehensive scanning and interpretation may take 10–15 min, which is less practical in high-throughput settings. Interpretation of the images from RCM necessitates specialized training [46]. The modality yields images similar to horizontal histologic sections, a convention to which many clinicians are not accustomed.

RCM continues to demonstrate considerable clinical usefulness. Its ability to increase diagnostic precision, inform treatment, and decrease invasiveness renders it a revolutionary aid in non-invasive imaging in dermatology.

### 2.4. Optical Coherence Tomography

OCT is a non-invasive imaging modality based on low-coherence interferometry that enables real-time, cross-sectional visualization of skin structures. Initially developed for ophthalmology, it has been adapted to dermatology, where it provides vertical images with a penetration depth of up to 1–2 mm and a resolution of nearly 10–15 µm laterally and nearly 5–10 µm axially, allowing assessment of both epidermal and dermal architecture [47]. Though OCT lacks cellular-level detail, its ability to capture the broader morphology of skin lesions makes it particularly useful in evaluating the deeper component of tumors, such as nodular basal cell carcinoma or invasive squamous cell carcinoma [48].

Typical OCT features of basal cell carcinoma include hyporeflective or hyper-reflective nests, peripheral hyporeflective clefts, and surrounding collagen hyper-reflectivity, while actinic keratoses may present as hyperkeratotic surface signals and disrupted epidermal layers [49]. These morphological features, though less specific than those of RCM, allow differentiation between benign and malignant lesions and support clinical decisions in ambiguous cases. A major innovation is the advent of Doppler-OCT (D-OCT), which adds the ability to detect and quantify blood flow in real time. By capturing motion-related changes in scattered light from red blood cells, D-OCT generates vascular maps superimposed on structural scans. This is especially useful in oncology, as malignant lesions often exhibit increased, irregular, or chaotic vasculature [d]. D-OCT has also demonstrated value in differentiating subtypes of BCC and melanoma based on vascular density, diameter, and branching patterns [50]. D-OCT shows promise in assessing inflammatory skin diseases, particularly in the scalp [51]. However, both OCT and D-OCT have limitations. The resolution is lower than that of RCM or LC-OCT, and the diagnostic accuracy is reduced in small or flat lesions. Furthermore, the lack of standard diagnostic criteria and limited operator experience may hinder widespread adoption [52]. D-OCT, while promising, remains largely confined to specialized centers and research settings.

In conclusion, OCT does give significant architectural information with its deeper imaging, providing an advantage for assessment of tumor invasion or subcutaneous infiltration. D-OCT adds functional vascular dimension to OCT that may increase diagnostic specificity according to vascular architecture. These combined techniques are part of a non-invasive, multimodal diagnostic methodology in dermatology and may potentially reduce the necessity of skin biopsy and enhance patient management.

### 2.5. Line-Field Confocal Optical Coherence Tomography

LC-OCT represents a significant leap forward in non-invasive skin imaging. By blending elements of confocal microscopy with the depth-resolving power of optical coherence tomography, this hybrid technique produces remarkably detailed in vivo images in both horizontal and vertical orientations, often described as offering “virtual histology” on the spot [53]. What sets LC-OCT apart is its use of a broadband light source in conjunction with line-field detection, allowing for cross-sectional scans that rival the clarity of confocal systems while penetrating roughly 500 μm into the skin, about twice the depth achievable with RCM. By directing a focused line of light onto the skin and capturing interference patterns, LC-OCT acquires en face (horizontal) and cross-sectional (vertical) images with resolutions of ~1.3 µm (horizontal) and ~1 µm (vertical). This dual-plane capability enables near-histologic 3D reconstructions of skin lesions (typically 1.2 × 1 × 0.5 mm) [54]. LC-OCT offers comparable epidermal and superficial dermal imaging to RCM, but with the added ability to visualize deeper tumor components. One study showed LC-OCT had superior diagnostic accuracy for skin carcinomas compared to clinical and dermoscopic assessment alone [55]. It can distinguish BCC subtypes (nodular, superficial, infiltrative) based on structural patterns and depth [56]. Early applications extend to inflammatory dermatoses: LC-OCT has detected features of cicatricial alopecia correlating with histology, and it shows potential for evaluating dermal fibrosis in conditions like morphea or lichen sclerosus [57]. The ability to co-localize dermoscopic features with LC-OCT images and provide real-time AI-assisted BCC predictions (Figure 6). LC-OCT makes it a compelling tool for pre-surgical tumor margin assessment [58]. Recently, LC-OCT is showing promising results in differential diagnosis between melanoma and nevi [59].

LC-OCT provides near-confocal resolution with greater depth, bridging the diagnostic gap between RCM and traditional OCT. It enables clear imaging of both superficial and deep skin structures. The dual modality allows comprehensive evaluation: horizontal views reveal architectural patterns, while vertical views display layering and depth of invasion. This facilitates intuitive correlation with standard histology and may reduce the diagnostic learning curve [60]. LC-OCT may significantly reduce the need for biopsies by providing diagnostic and subtype information in real time. Devices are being developed with AI algorithms that quantify the probability of BCC and provide differential diagnoses of actinic keratosis, squamous cell carcinoma, dermal nevus, sebaceous hyperplasia, melanocytic lesions, and healthy skin [61].

LC-OCT is a promising non-invasive diagnostic tool that offers real-time, near-histologic visualization of skin lesions. It holds great potential for enhancing diagnostic accuracy, guiding management decisions, and reducing the need for invasive procedures. As research and clinical experience grow, LC-OCT may emerge as a transformative tool in dermatology, redefining non-invasive diagnosis.

### 2.6. Emerging Technologies and Artificial Intelligence-Based Tools

The implementation of three-dimensional (3D) digital imaging systems combined with artificial intelligence (AI) has significantly transformed the non-invasive diagnosis of skin cancers, particularly melanoma, offering new avenues for both early detection and long-term surveillance. Among these technologies, the Vectra WB360 system (Canfield Scientific) and the European project Intelligent Total Body Scanner for Early Detection of Melanoma (iToBoS) represent highly advanced platforms that integrate full-body imaging, digital dermoscopy, machine learning algorithms, and multimodal data fusion to support personalized melanoma management.

The Vectra WB360 utilizes an array of 92 synchronized cameras arranged circumferentially to capture high-resolution full-body 3D images in seconds without any physical contact (Figure 7). The rapid acquisition enables clinicians to obtain a comprehensive map of the patient’s skin surface, providing detailed anatomical models that can be longitudinally compared to monitor morphological changes over time. Integrated dermoscopic modules allow synchronous capture of high-resolution dermoscopic images for selected lesions, which are then linked to their precise 3D body location, enhancing both lesion tracking and risk assessment across follow-up sessions [62].

This combination of 3D total body photography with linked dermoscopic imaging has demonstrated substantial improvements in diagnostic performance in high-risk populations. In a real-world prospective cohort using convolutional neural networks (CNNs) trained on 3D datasets, sensitivity for melanoma detection reached 94.1%, with specificity approaching 88.1% [62]. This represents a clinically significant advancement, particularly for patients with multiple nevi, personal or family history of melanoma, or phenotypic risk factors that complicate visual assessment.

An additional innovation in the Canfield Scientific platform is the integration of the DEXI (Dermoscopy Explainable Intelligence) cognitive assistant, which enhances clinical interpretation of dermoscopic images by providing automated risk assessment and visual analytic support. DEXI generates lesion-specific scores based on key morphometric parameters—such as asymmetry, border irregularity, color variation, and diameter—and displays similarity matches with previously classified lesions from curated image databases. This decision-support tool is particularly valuable in high-risk populations, facilitating lesion prioritization and complementing visual examination during longitudinal digital monitoring (Figure 8).

The iToBoS platform, developed by a multidisciplinary European consortium under the Horizon 2020 program, represents a potential next generation in total body imaging technologies by integrating diverse optical components and artificial intelligence (AI)-based data analysis workflows [63] (Figure 9). The system acquires multiple high-resolution two-dimensional (2D) images from different angles, which are subsequently registered and geometrically integrated to generate a superficial three-dimensional (3D) anatomical model of the patient’s skin. This model is used for anatomical mapping, precise lesion localization, and longitudinal monitoring [64]. Cameras equipped with liquid lenses allow for rapid focus adjustments, while multispectral illumination and polarization filters enhance subsurface visualization. Focus stacking and super-resolution algorithms further optimize image quality, enabling structural representations comparable to those of contact dermoscopy [63].

Beyond image acquisition, one of the main innovations of iToBoS lies in its AI-enhanced analytical capabilities. The system generates a longitudinal dataset containing both 3D total body surface models and high-resolution dermoscopic-like images, which are used to train various algorithms capable of automatically detecting lesions, segmenting them, classifying them morphologically, and assigning dynamic risk scores [64,65] (Figure 10 and Figure 11). These algorithms consider diverse clinical and biological variables, including lesion location, chronological morphological evolution, observed dermoscopic structures, and patient-specific factors such as Fitzpatrick phototype, age, family history of melanoma, or genetic predisposition, in order to generate individualized risk profiles.

This automated lesion prioritization process could facilitate clinical decision-making by highlighting lesions that exhibit suspicious temporal changes or a higher probability of malignancy. In addition to potentially improving early melanoma detection, these technologies could reduce unnecessary excisions of benign nevi, optimize clinical workflows, and enhance the allocation of dermatological resources in both tertiary hospital settings and population screening programs [63,64].

The performance of AI models in skin cancer classification has been extensively validated across multiple independent datasets. Various convolutional neural network architectures trained on large dermoscopic databases (such as the ISIC archive) have consistently demonstrated diagnostic accuracy comparable to or greater than that of expert dermatologists, with area under the curve (AUC) values exceeding 0.91 for melanoma detection. However, despite this promising performance in controlled research contexts, broader clinical implementation would require external multicenter validations, workflow harmonization, continuous medical oversight, and doctor–patient communication protocols to mitigate the risks of false negatives or positives [65].

The integration of 3D total body imaging with AI-powered analytics would offer unique advantages over traditional models focused on single-lesion evaluation. By enabling complete skin mapping, longitudinal monitoring of multiple lesions, and objective quantification of their dynamics over time, these platforms would provide a multidimensional diagnostic approach that accounts for the heterogeneity and evolving behavior of melanocytic lesions. This would be especially valuable in patients with numerous nevi, dysplastic nevi, or familial melanoma syndromes, where distinguishing between stable nevi and emerging melanomas remains a constant clinical challenge.

Moreover, these systems would contribute valuable datasets to the continuous training of AI algorithms, generating a feedback loop in which each successive scan enriches the algorithm’s knowledge base with real clinical variability, thereby improving its robustness and generalizability [66,67,68,69]. The incorporation of explainable artificial intelligence (XAI) algorithms could further enhance clinical acceptance by allowing professionals to visualize the AI’s reasoning process, fostering transparency and facilitating multidisciplinary adoption in routine dermatologic practice [69,70,71].

In summary, the convergence of contactless 3D total body photography, high-resolution digital dermoscopy, longitudinal image databases, and multimodal AI algorithms would represent a paradigm shift in melanoma screening and monitoring. These technologies could support earlier diagnosis, individualized monitoring, and more efficient allocation of dermatologic resources while maintaining clinical oversight and enhancing patient safety. Prospective clinical validation, continuous technological refinement, and structured integration into routine care will be essential to realize their transformative potential in global melanoma management.

### 2.7. Applications of Mobile Apps, Electrical Impedance Spectroscopy (EIS), and Multispectral Imaging in Dermatology

Mobile applications utilizing AI have emerged as accessible tools in dermatologic diagnostics, primarily aimed at early detection and monitoring of skin lesions, especially melanoma. These apps typically use smartphone cameras and algorithms trained on extensive image databases to assess skin lesions and provide diagnostic suggestions. A large prospective study of the CE-certified SkinVision^®^ app involving 1204 pigmented lesions showed that the algorithm labelled 27 times more lesions as “high risk” than board-certified dermatologists and achieved only modest performance (AUC 0.62–0.72; sensitivity 41–83%, specificity 60–83%), illustrating the problem of over-detection and the downstream burden of unnecessary excisions [72]. Likewise, the MEL-SELF pilot randomized controlled trial of patient-led smartphone teledermoscopy found that clinicians valued improved access and earlier reassurance but warned that variable image quality and medicolegal uncertainty could still provoke medical overuse unless clear triage pathways are built around app output [73]. A study highlighted several limitations, including inadequate supporting evidence, absence of clinician oversight, opaque algorithm development processes, questionable data handling, and insufficient privacy protection. Of 41 apps analyzed, none had FDA approval, only a few included clinician input, and nearly half did not clearly specify data privacy measures, raising concerns about patient safety, privacy, and clinical reliability [74]. A real-world evaluation conducted in Italy using a deep convolutional neural network (CNN) integrated into a mobile app (“Clicca il Neo”) demonstrated moderate diagnostic accuracy (AUC of 0.67), highlighting the gap between experimental and practical clinical utility [75]. A recent study involving 16 dermatologists and 17 general practitioners in the Netherlands explored why mHealth triage apps remain marginal in routine care. While participants acknowledged three advantages—heightened public vigilance, quicker flagging of suspicious lesions, and reduced unnecessary visits—they also voiced five core concerns: unreliable outputs; algorithmic bias that disadvantages darker skin tones or digitally naïve users; erosion of GP autonomy; murky legal accountability; and rising medico-legal costs when decisions go wrong. The participants concluded that endorsement hinges on four safeguards: independently audited accuracy studies on representative community cohorts; seamless integration of app reports into electronic health records; nationally agreed liability rules that clarify responsibility for missed malignancies; and inclusive, privacy-by-design interfaces that guide image capture and flag poor photo quality. Until these benchmarks are met, the clinicians stated that they would still recommend a conventional face-to-face assessment over a stand-alone consumer app for most members of the study panels [76]. The major limitations of mobile dermatologic apps include potential anxiety induction from false-positive results and inappropriate patient reassurance in false negatives. Clear guidelines and ongoing clinician involvement are considered essential to ensure diagnostic accuracy and appropriate patient counseling.

#### 2.7.1. Electrical Impedance Spectroscopy (EIS)

EIS is a non-invasive technique assessing skin impedance through electrical currents. Malignant and benign tissues exhibit distinct impedance patterns due to structural differences. EIS is valuable in diagnosing melanoma, showing sensitivities of 92–98% and specificities of 50–80% [77]. EIS, particularly devices like Nevisense, complements clinical and dermoscopic findings, especially in ambiguous cases [78]. While effective, EIS limitations include dependency on lesion characteristics (size, ulceration, location) affecting measurement accuracy [79]. Moreover, interpretation requires specialized training, and availability remains limited outside specialized settings. Zakria et al. invited dermatologists to review images of 49 histologically confirmed melanomas, severe dysplastic nevi, and benign pigmented lesions in a three-step exercise: (i) clinical photograph, (ii) clinical + dermoscopy, and (iii) clinical + dermoscopy supplemented with a blinded EIS score. Of the 33,957 biopsy decisions made, dermoscopy significantly improved the selection of correct biopsies compared to using the clinical image alone. Adding the EIS score provided an additional, statistically significant improvement for all lesion categories, while also reducing the number of unnecessary biopsies of benign lesions [80]. These improvements were evident irrespective of clinician seniority or practice environment, demonstrating the potential of EIS to enhance decision-making when morphology alone is inconclusive. Integrating EIS with teledermoscopy has shown improved diagnostic accuracy, particularly beneficial in remote or underserved communities [81].

#### 2.7.2. Multispectral Imaging

Multispectral imaging uses multiple wavelengths, extending beyond visible light to near-infrared, enhancing the visualization of lesion features. Multispectral imaging significantly augments diagnostic capabilities, achieving sensitivities of 90–95% and specificities of 85–90% [82,83]. These imaging technologies, particularly when integrated with deep learning techniques, enhance early detection and diagnostic accuracy, offering non-invasive alternatives to traditional biopsy methods [83,84,85]. Despite promising results, multispectral imaging’s widespread adoption faces barriers including high costs, required specialized equipment, and training. The risk of false positives necessitates correlation with clinical findings to optimize diagnostic accuracy. While mobile apps, EIS, and multispectral imaging represent significant advances in dermatologic diagnostics, their clinical utility is contingent on robust validation, transparency, clinician oversight, and appropriate patient education. Enhancing these elements will be crucial to maximizing their potential, minimizing risks, and improving overall patient outcomes in dermatology.

## 3. Discussion

Traditional dermatological diagnostic tools such as the Tzanck smear, KOH microscopy, scabies preparation, and Wood’s lamp examination remain valuable for their accessibility and rapid results. As already said, the Tzanck smear provides a quick cytological examination to identify viral infections such as herpes simplex and varicella-zoster, but also autoimmune conditions like pemphigus vulgaris, though it lacks the specificity to differentiate between virus types. Despite molecular techniques such as PCR being superior in sensitivity and specificity, the Tzanck smear continues to be essential, particularly in resource-limited settings.

Similarly, KOH microscopy is fundamental in diagnosing superficial fungal infections, offering high sensitivity (80–95%) and specificity (>90%). It is complemented by Wood’s lamp examination, which utilizes ultraviolet fluorescence to identify certain dermatophytoses, erythrasma, and pigmentary disorders like vitiligo, reinforcing its relevance in initial clinical evaluations. Scabies microscopy, while variable in sensitivity (40–90%), remains crucial when clinical suspicion is high, although negative findings should prompt further investigation rather than exclusion.

Dermoscopy significantly transformed dermatological diagnostics by enabling detailed examination of subsurface skin structures. While conventional dermoscopy provides high diagnostic accuracy for melanoma (83–89% sensitivity) and basal cell carcinoma (92–96% sensitivity), its effectiveness is contingent on clinician expertise. The evolution towards digital dermoscopy, integrating advanced quantitative analysis and deep learning, has augmented diagnostic precision and longitudinal lesion monitoring, enhancing early detection capabilities, especially crucial in high-risk melanoma patients.

Further advancements such as optical super-high magnification dermoscopy (up to 400× magnification) and UV-induced fluorescence dermoscopy have expanded diagnostic possibilities, improving visualization of cellular structures and microbial agents. These modalities enhance diagnostic differentiation, particularly beneficial in complex neoplastic and inflammatory conditions.

RCM offers cellular-level resolution, closely approximating histological examination without invasive biopsy. RCM’s high sensitivity (92–97%) and specificity (70–80%) for melanoma and over 90% for basal cell carcinoma highlight its potential in reducing unnecessary biopsies. Nevertheless, limitations such as restricted penetration depth (250–300 µm) and time-intensive mosaicking procedures highlight its use primarily for ambiguous lesions and in specialized clinical settings.

LC-OCT combines dermoscopy with RCM to deliver deeper penetration of 500 µm and dual-plane imaging which boosts diagnostic accuracy for skin carcinomas and inflammatory disorders while providing straightforward correlations with histological structures. The promising technology faces barriers to clinical adoption because of its limited use in practice, time-consuming image acquisition process, and requirement for specialized training.

The third-level diagnostic tools Vec-tra WB360 and iToBoS use AI technology to provide full-body imaging systems that combine digital dermoscopy with three-dimensional modeling and advanced algorithms for both melanoma detection and long-term patient monitoring. The technologies show high sensitivity rates of up to 94% yet their implementation requires thorough external testing and medical supervision to maintain reliable diagnoses and prevent incorrect positive or negative results.

The field of dermatological diagnostics experiences expansion through new technologies which include mobile dermatologic apps and Electrical Impedance Spectroscopy (EIS) and multispectral imaging. Mobile applications that use artificial intelligence to deliver preliminary assessments need thorough validation processes and strict privacy standards for implementation. The sensitivity of EIS for melanoma detection is high but its specificity needs improvement while multispectral imaging improves diagnostic accuracy yet faces implementation challenges because of its expensive nature and complex setup.

The diagnostic process for dermatology needs to use a systematic Bayesian-based method which selects tests according to the level of clinical doubt. The first-level tests serve as essential diagnostic tools when suspicion is high because they enable quick diagnosis confirmation. RCM and LC-OCT tools used in second-level diagnostics improve ambiguous case diagnosis accuracy but third-level AI-based advanced imaging systems require extensive validation before they can deliver improved early detection and management results to clinical practice.

## 4. Conclusions

Nowadays several non-invasive diagnostic tools are available to dermatologists, both traditional, more simple, low-cost tests and more complex imaging options with greater sensitivity and specificity. An ideal diagnostic approach relies on intelligent choices among these instruments associated with clinical suspicion and under overseas by Bayesian logic (Figure 12). In this context, the worth of a diagnostic test is not exclusively determined by its own inherent sensitivity and specificity; what matters is also the pre-test probability of the disease, which significantly impacts the post-test probability of the disease.

Tests with poor diagnostic accuracy (sensitivity and specificity both around 50%) have little practical use; they do not change the likelihood of disease regardless of the outcome. In such a situation, the results of tests are not likely to affect clinical decision-making to a great extent. Conversely, with a very low pre-test probability (<5%) or a very high pre-test probability (>90%), the effect of testing is small. In such circumstances, particularly when the test is expensive or time-consuming, its routine application might not be justified.

On the other hand, diagnostic tests have their highest value in situations where there is truly uncertainty, which means usually when the clinician is considering two or three different diagnoses, and where pre-test probabilities are more in the 33–50% range. In this in-between setting, tests with sensitivity and specificity can markedly shift the post-test probability, improving the diagnostic process helping the dermatologist in daily clinical practice.

## Figures and Tables

**Figure 1 diagnostics-15-02100-f001:**
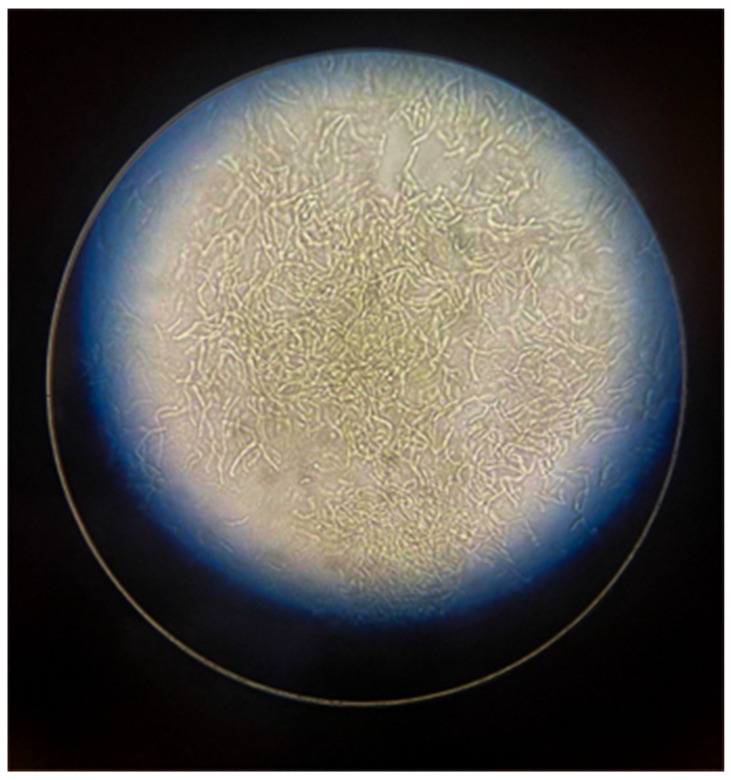
An example of preparation with KOH for the detection of filamentous fungi.

**Figure 2 diagnostics-15-02100-f002:**
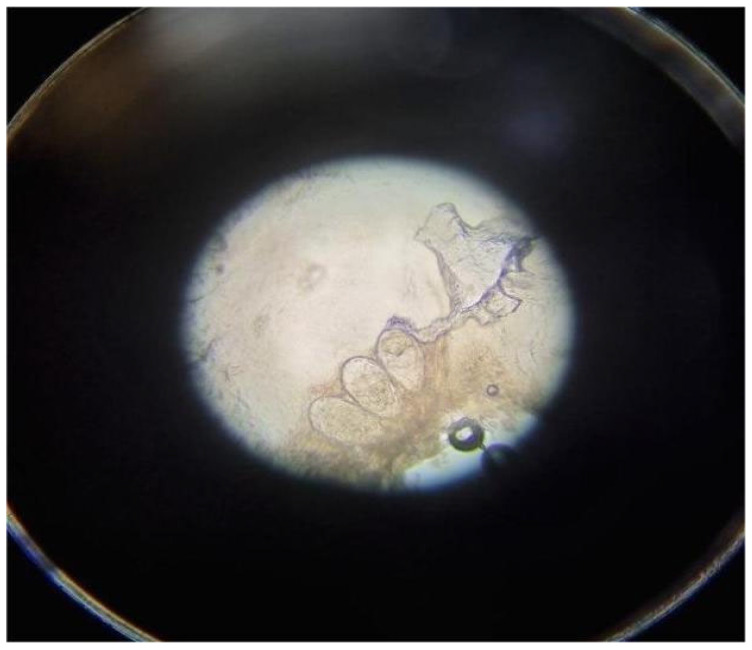
Microscopic examination with potassium hydroxide (KOH) preparation for the detection of eggs of *Sarcoptes scabiei*.

**Figure 3 diagnostics-15-02100-f003:**
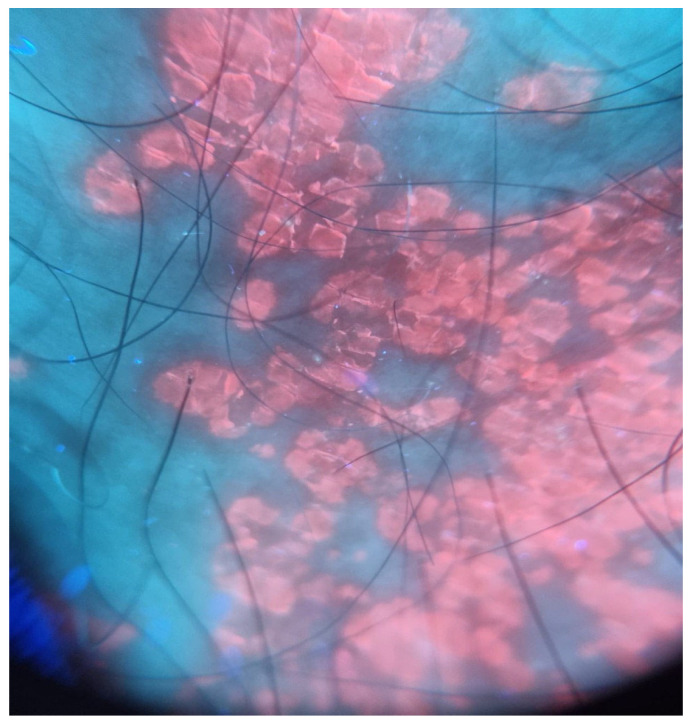
Erythrasma under Wood’s light.

**Figure 4 diagnostics-15-02100-f004:**
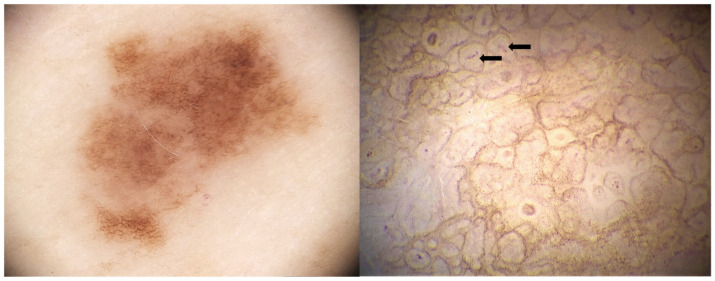
**Left**: Atypical melanocytic nevus visualized with 20× dermoscopy using the Medicam 1000s camera and D-Scope IV lens (Fotofinder System). An atypical pigment network with some globules is observed. **Right**: With optical super high magnification dermoscopy (OSHMD) using the D-Scope III lens and the same camera (Fotofinder System, Bad Birnbach, Germany), the pigment network appears as in-focus brown rings composed of small round or polygonal brown structures surrounding dermal papillae; within the papillae, small capillary loops can be seen (black arrows).

**Figure 5 diagnostics-15-02100-f005:**
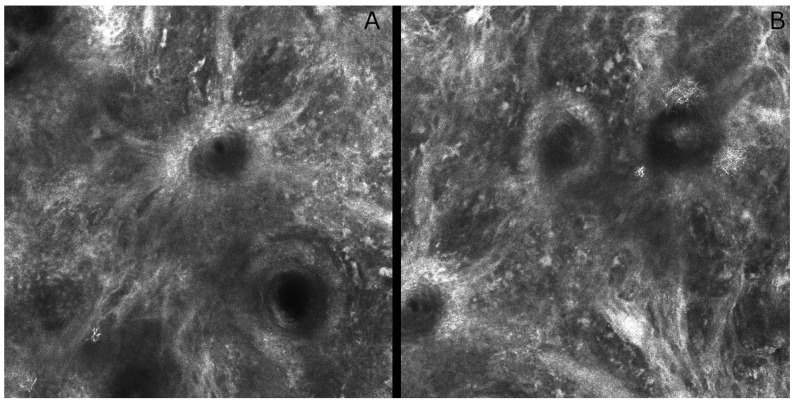
Reflectance confocal microscopy images of a lentigo maligna. Elongated nest of a lentigo maligna with folliculotrophophic hyper-reflective atypical cells forming a “caput medusae” structure (**A**). Atypical diffuse hyperreflective melanoma cells forming elongated nests and invading the dermo-epidermal junction, which appears blurred and disrupted due to the downward migration of hyper-reflective atypical cells into the dermis, leading to a non-edged papillae pattern (**B**). (field of view of 500 × 500 μm).

**Figure 6 diagnostics-15-02100-f006:**
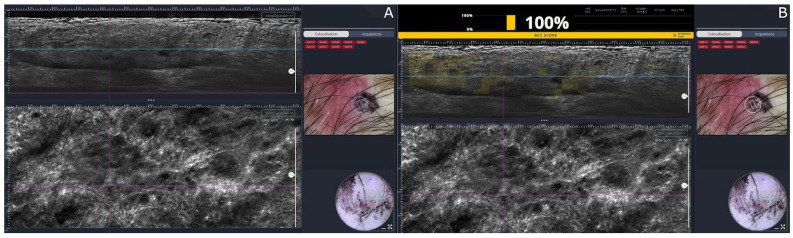
LC-OCT: Superficial basal cells carcinoma under line-field confocal optical coherence tomography. In the left image a superficial basal cells carcinoma is seen with a millefouille pattern and clefting at the periphery (**A**). In the right image the same BCC is shown with the artificial intelligence assistant BCC prediction scoring a 100% probability of BCC (**B**).

**Figure 7 diagnostics-15-02100-f007:**
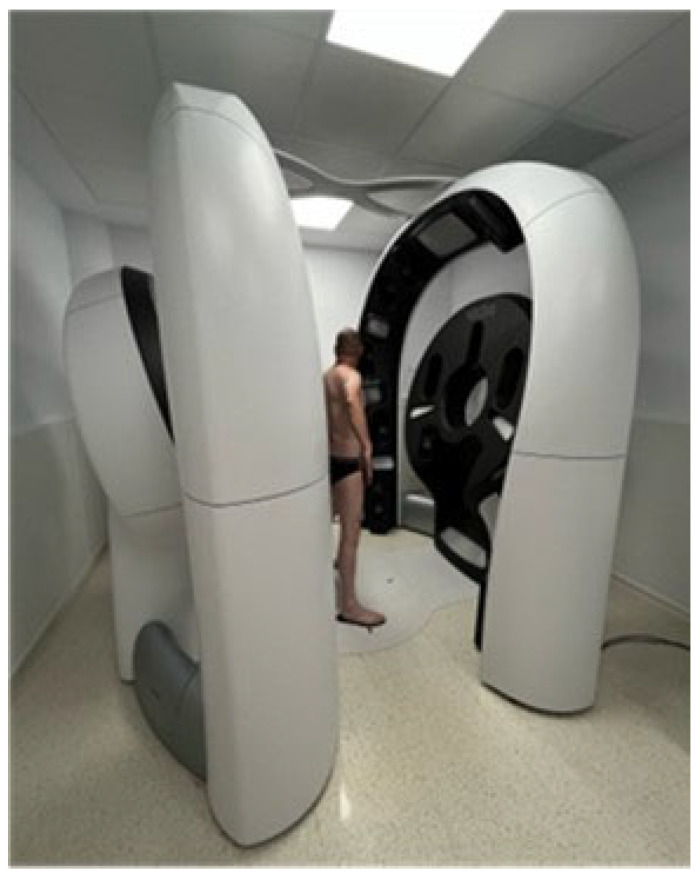
Vectra WB360 (Canfield Scientific): automated 3D full-body imaging system integrating synchronized cameras and linked dermoscopy modules for contactless skin surface acquisition and longitudinal lesion monitoring.

**Figure 8 diagnostics-15-02100-f008:**
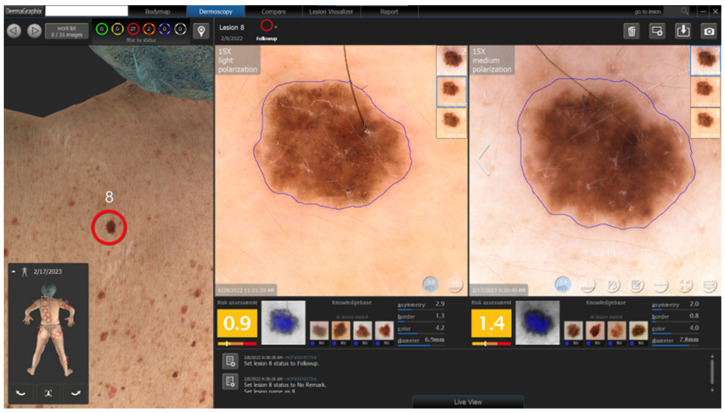
AI-assisted dermoscopic follow-up using the D200 system (Canfield Scientific) with integration of the DEXI cognitive assistant. Automated risk scores, morphometric parameters (asymmetry, border, color, and diameter), and lesion similarity matching from reference databases are displayed. This tool complements clinical visual assessment, supporting diagnostic prioritization in high-risk settings.

**Figure 9 diagnostics-15-02100-f009:**
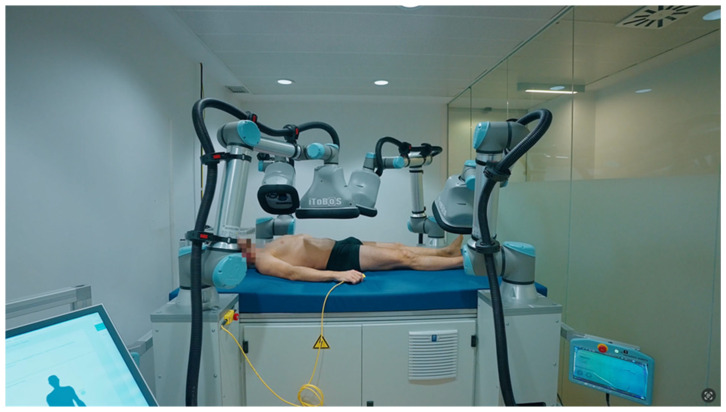
iToBoS prototype: European Horizon 2020 Intelligent Total Body Scanner platform integrating multiple high-resolution 2D cameras with liquid lenses, multispectral illumination, and AI-based data fusion to reconstruct 3D anatomical models for melanoma risk assessment.

**Figure 10 diagnostics-15-02100-f010:**
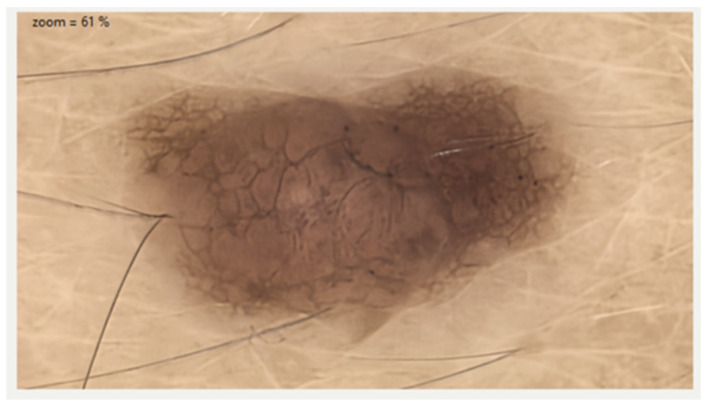
Non-contact dermoscopic image acquired with the iToBoS system. A melanocytic lesion is observed, showing an atypical pigment network and irregularly contoured areas of hyperpigmentation. The system uses liquid lenses and super-resolution algorithms to optimize morphological detail without requiring direct skin contact.

**Figure 11 diagnostics-15-02100-f011:**
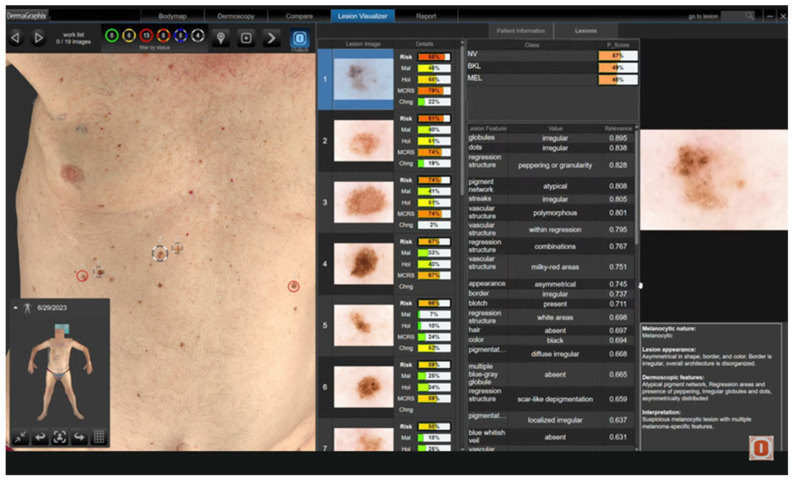
Graphical interface of the iToBoS system showing lesion prioritization through artificial intelligence. The platform integrates 2D body images with automated analysis, including risk scores, malignancy probability, morphological changes, and dermoscopic structure classification. It enables longitudinal follow-up and generates individualized risk assessments to support clinical decision-making.

**Figure 12 diagnostics-15-02100-f012:**
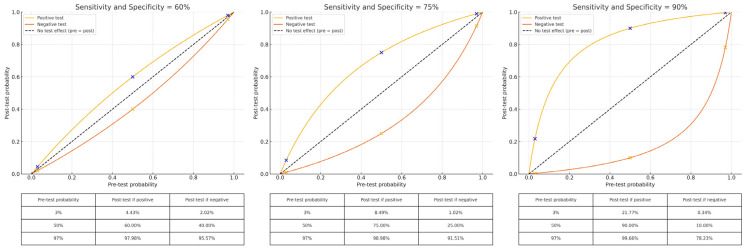
Impact of diagnostic test accuracy and pre-test probability on post-test outcomes in dermatology. This composite image illustrates how diagnostic test performance influences post-test probability across different levels of pre-test probability, according to Bayes’ theorem. The three panels represent tests with increasing accuracy: sensitivity and specificity of 60%, 75%, and 90%. The dashed diagonal line marks the identity line, where the post-test probability equals the pre-test probability, indicating that the test result does not alter diagnostic confidence. The blue and orange curves depict the post-test probability when the test results are positive and negative, respectively. Superimposed on each curve are three points, represented by blue Xs on the positive-result curve and orange Xs on the negative-result curve, corresponding to pre-test probabilities of 3%, 50%, and 97%. These illustrate how the diagnostic yield of a test varies across different clinical contexts. When test accuracy is low, or when the pre-test probability is extremely low or high, the result has little impact on decision-making. In contrast, high-performance tests applied in settings of diagnostic uncertainty (pre-test probability around 30–50%) significantly shift the post-test probability, thereby enhancing diagnostic precision and clinical utility.

## Data Availability

Not applicable.

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
