# Peer review of "Advancements in Diagnosis of Neoplastic and Inflammatory Skin Diseases: Old and Emerging Approaches"

_diagnostics, 2025, doi:10.3390/diagnostics15162100_

Round 1

Reviewer 1 Report

Comments and Suggestions for Authors

The authors provide a comprehensive overview of both traditional and modern diagnostic techniques in dermatology, discussing methods ranging from KOH examination to more recent non-invasive tools such as RCM and EIS, along with their diagnostic accuracy. The manuscript is very well organized; however, the following points should be addressed:

  1. Figure 1 appears overly bright. Please adjust the brightness and contrast so that the fungal elements are more clearly visible.
  2. Figure 2 shows only the eggs of scabies. Please explicitly state that these are eggs.
  3. Line 194: It would be beneficial to cite a reference regarding the appearance of vitiligo under Wood’s lamp.
  4. Figure 5 lacks an indication of scale; please include either the magnification or a scale bar. Is this a horizontal section? On what basis do you identify the dermo-epidermal junction in this image? Clarifying this point would likely enhance readers’ interest in the technique.

.

Author Response

Reviewer n’1: Figure 1 appears overly bright. Please adjust the brightness and contrast so that the fungal elements are more clearly visible.

Answer n’1: Done, thank you.

Reviewer n’1: Figure 2 shows only the eggs of scabies. Please explicitly state that these are eggs.

Answer n’2: Done, thank you.

Reviewer n’1: Line 194: It would be beneficial to cite a reference regarding the appearance of vitiligo under Wood’s lamp.

Answer n’3: Done, thank you.

Reviewer n’1: Figure 5 lacks an indication of scale; please include either the magnification or a scale bar. Is this a horizontal section? On what basis do you identify the dermo-epidermal junction in this image? Clarifying this point would likely enhance readers’ interest in the technique.

Answer n’4: Thank you for your valuable comment.

We have now included a scale bar in Figure 5, indicating a field of view of 500 × 500 μm.

Regarding the identification of the dermoepidermal junction, in this specific case, the epidermis is composed of atypical, hyperreflective melanocytic cells, while the dermis appears hyporeflective, interspersed with some hyperreflective collagen strands. In healthy skin, the junction is typically well demarcated, characterized by a clear transition between the hyperreflective epidermis and the hyporeflective dermis, with distinct papillae edged architecture.

However, in this lesion, as commonly observed in invasive neoplasms, the junction appears blurred and disrupted, due to the downward migration of hyperreflective atypical cells into the dermis, leading to a non edged papillae pattern. This loss of definition supports the interpretation of dermal invasion, consistent with the diagnosis.

We hope this clarification improves the understanding and interest in the imaging technique.

Thank you.

Reviewer 2 Report

Comments and Suggestions for Authors

Federico et al present in this narrative review diagnostic approaches for neiplasitc and inflammatory skin diseases including classical and modern methods.

Following comments:

The paper is focussing more on neoplastic than inflammatory diseases. The title is therefore somewhat misleading.

In part is reads as advertisement for the presented techniques. Is there any plagiarism?

Figures 1 and 2 are hardly to read and should be substited by better photographs. Pleases include arrows to indicate relevant structures.

Page 7, line254: please explain „Jet with contrail“.

Page 9, line298: clefts in basal cell carcinoma have for long been regarded are a histological artifact, yet due to local mucin deposition (?) seem to really exist in vivo as well. This should be briefly commented.

Page 13, line 427: please explan DEXI.

Author Response

Reviewer n’2: The paper is focussing more on neoplastic than inflammatory diseases. The title is therefore somewhat misleading.

In part is reads as advertisement for the presented techniques. Is there any plagiarism?

Answer n’1: Dear Reviewer n’2, thank you very much for your valuable comment. So, we think that also the inflammatory dermatology is covered by our review, but we have put the “neoplastic” as first  because we are focused mostly on neoplasms. Thank you.

There is no plagiarism because we conduct only a narrative review of the old and modern technique.

Thank you.

Reviewer n’2: Figures 1 and 2 are hardly to read and should be substited by better photographs. Pleases include arrows to indicate relevant structures.

Answer n’2: Thank you. We have adjusted the contrast of the figure 1 in which is possible now to see the fungal elements. We think that now it’s simpler to see. We are not able to have another original picture of eggs of Sarcoptes Scabiei. Sorry about that. Thank you.

Reviewer n’2: Page 7, line254: please explain „Jet with contrail“.

Answer n’3:  "Jet with contrail" is the evident burrow ending with a triangular brownish projection, corresponding to the mite.

Reviewer n’2: Page 9, line298: clefts in basal cell carcinoma have for long been regarded are a histological artifact, yet due to local mucin deposition (?) seem to really exist in vivo as well. This should be briefly commented.

Answer n’4: Thank you for your insightful comment. We have added a clarification in the revised manuscript to address this point. While clefts in basal cell carcinoma (BCC) were historically considered a fixation-related histological artifact, recent studies using in vivo reflectance confocal microscopy (RCM) and optical coherence tomography (OCT) have shown that these structures actually correspond to true peritumoral spaces filled with mucin. Ulrich et al. (2011) first demonstrated a strong correlation between hypo-reflective clefts seen on RCM and mucin deposits observed histologically using Alcian blue staining. This finding has been consistently confirmed in a more recent work Lupu et al. (2023), who supports the interpretation that peritumoral clefts represent real stromal changes due to mucin accumulation, rather than mere processing artifacts. A brief explanatory sentence has been added to the text accordingly (Page 9, Line 298).

References:

Peritumoral clefting in basal cell carcinoma: correlation of in vivo reflectance confocal microscopy and routine histology – Ulrich M., Roewert-Huber J., González S., Rius-Díaz F., Stockfleth E., Kanitakis J. (2011), Journal of Cutaneous Pathology, DOI: 10.1111/j.1600-0560.2010.01632.x

In Vivo Characterization of Mucin and Amyloid Deposits in Primary Basal Cell Carcinoma through Reflectance Confocal Microscopy: A Correlation with Histopathology – Lupu M., Malciu A.M., Voiculescu V.M. (2023), Diagnostics, DOI: 10.3390/diagnostics13030422

Reviewer n’2: Page 13, line 427: please explan DEXI.

Answer n’5: Done. Thank you.

Round 2

Reviewer 1 Report

Comments and Suggestions for Authors

I have no more comment.

Reviewer 2 Report

Comments and Suggestions for Authors

all major comments have been followed